# Targeted and Non-Targeted HPLC Analysis of Coffee-Based Products as Effective Tools for Evaluating the Coffee Authenticity

**DOI:** 10.3390/molecules27217419

**Published:** 2022-11-01

**Authors:** Jitka Klikarová, Lenka Česlová

**Affiliations:** Department of Analytical Chemistry, Faculty of Chemical Technology, University of Pardubice, Studentská 573, CZ-53210 Pardubice, Czech Republic

**Keywords:** coffee, targeted analysis, non-targeted fingerprint, HPLC

## Abstract

Coffee is a very popular beverage worldwide. However, its composition and characteristics are affected by a number of factors, such as geographical and botanical origin, harvesting and roasting conditions, and brewing method used. As coffee consumption rises, the demands on its high quality and authenticity naturally grows as well. Unfortunately, at the same time, various tricks of coffee adulteration occur more frequently, with the intention of quick economic profit. Many analytical methods have already been developed to verify the coffee authenticity, in which the high-performance liquid chromatography (HPLC) plays a crucial role, especially thanks to its high selectivity and sensitivity. Thus, this review summarizes the results of targeted and non-targeted HPLC analysis of coffee-based products over the last 10 years as an effective tool for determining coffee composition, which can help to reveal potential forgeries and non-compliance with good manufacturing practice, and subsequently protects consumers from buying overpriced low-quality product. The advantages and drawbacks of the targeted analysis are specified and contrasted with those of the non-targeted HPLC fingerprints, which simply consider the chemical profile of the sample, regardless of the determination of individual compounds present.

## 1. Introduction

The coffee tree belongs to the Coffea genus of the Rubiaceae family, including more than 100 species, of which Coffea Arabica (Arabica) and Coffea Canephora (Robusta) are the most consumed, and therefore the most economically important [1,2,3]. Arabica differs from Robusta in several aspects, such as morphology, size and colour of the beans, chemical composition, and sensory properties [4,5,6], as well as growing, cultivation, and brewing properties [7]. Robusta provides very good body and foam, is richer in chlorogenic acids, and contains approximately 40–50% more caffeine than Arabica, which accounts for 65% of global production, is more acidic, less bitter, and has a more refined and pronounced taste and aroma [7,8,9,10,11]. For this reason, Arabica is much more appreciated by coffee consumers, hence its market price is approximately 20–25% higher compared to Robusta [12].

The coffee tree is grown in about 60 countries around the world, for which it is a crucial economic item [13,14]. Coffee beans have been considered one of the most important food products, playing various roles in economic, political, and religious issues since time immemorial [15]. Their total global production is an incredible 9.7 million tonnes per year [16], with approximately 70% of coffee production coming from only three countries, namely Brazil, Vietnam, and Colombia [2,17]. The production of high-quality coffee beans does not only depend on harvest and post-harvest operations, but also requires the right choice of growing area of individual cultivars (climate, soil, altitude, etc.) and proper planting and storage of the beans [18,19,20].

The phytochemical profile of green coffee beans is currently known to be very complex and provides a wide range of health benefits [21,22]. Coffee has been valued for years for its stimulating effect on the central nervous system, associated primarily with caffeine [23,24,25]. Nevertheless, studies show that consumption of two to three coffee cups a day brings many other potential health benefits, including prevention of cancer, type 2 diabetes, cardiovascular and liver diseases, and Alzheimer’s and Parkinson’s diseases [24,25,26,27,28,29,30,31,32,33,34]. In addition to caffeine, the most important bioactive compounds responsible for these effects are mainly polyphenols [21,35,36,37], of which esters of caffeic and quinic acids, known as chlorogenic acid isomers, are the most abundant [23,24]. While caffeic acid has anticancer effects [38], chlorogenic acids (CGA, Figure 1), including the isomers of caffeoylquinic (CafQA), dicaffeoylquinic (diCafQA), feruloylquinic (FQA), and p-coumaroylquinic (pCoQA) acids, exhibit antibacterial, antifungal, antiviral, antioxidant, and chemoprotective properties [34,39,40]. Coffee polyphenols, together with caffeine, also balance cholesterol and arrhythmia, reduce lipid oxidation and risk of obesity, hypertension, hyperglycemia, or heart and liver failure [30,41,42,43,44,45]. However, caffeine is also associated with stomach irritation, insomnia, and increased breathing and heart rate [34]. As regards the organoleptic characteristics of the coffee beverage, polyphenols are considered to be responsible for its acidity, bitterness, and astringency [46,47].

The typical organoleptic properties of coffee arise just during the roasting of green coffee beans. In the standard roasting process, temperature and time range between 180–250 °C and 2–25 min, respectively, depend on the required degree of roasting and the technique used [48]. Roasting is a very complex process during which countless chemical reactions occur (e.g., Maillard and Strecker reactions, followed by epimerization, decarboxylation, lactonization, and dehydration), which fundamentally change the chemical composition of the coffee beans (e.g., an alteration in the concentration of specific molecules and/or a formation of new and absolutely different ones), and thus also the taste, texture, and aroma of the coffee cup [3,11,49,50,51,52]. The Maillard reaction, i.e., the reaction between reducing sugars and free amino acids or peptides occurring at high temperatures, gives rise to an important class of brown polymeric compounds called melanoidins, which contribute to the typical colour, characteristic aroma, and pleasant bitterness of coffee beans [46,53,54]. Thermal decomposition of carbohydrates also leads to the formation of 5-hydroxymethylfurfural (5-HMF), which is an indicator of coffee deterioration caused by excessive roasting time and/or temperature or long storage of coffee [18,55]. Although 5-HMF has carcinogenic properties, it poses no risk to consumers due to its innocuous amount in coffee [56].

The coffee beverage is prepared by the infusion of roasted ground coffee seeds and, due to its organoleptic properties and stimulating effects, is consumed by millions of people worldwide every day. Coffee is currently one of the most consumed food products, and thus has become part of our everyday culture [1,15]. While flavour, described as a balanced combination of body, aroma, and taste without any defects, is essential parameter for the ordinary consumers [57], expert cuppers trained in accredited labs evaluate the quality properties of the coffee from several factors (based on a scoring scale developed by the American Specialized Coffee Association). They focus not only on fragrance of the coffee powder and aroma, intensity, acidity, bitterness, astringency, and body of the final beverage, but also on the presence of defective coffee beans including green (unripe), burnt, black (caused by its fermentation and/or delayed harvests), black-green (beans with adhered silver film caused by their drying at high temperature), broken or insect-damaged beans (especially due to the coffee borer beetle), as well as the occurrence of potential contaminants, such as plant debris, rocks, clods, or sticks [58,59].

Green coffee beans consist mainly of carbohydrates, which occupy up to 60% of the dry matter. Lipids (8–18%) and proteins (9–16%), including peptides and amino acids, are present in lower amounts [33]. It is generally claimed that coffee contains substances from more than 1000 different chemical classes (1500 chemicals, 850 volatiles, and 700 dissolved compounds). However, their precise representation and quantity depend on many variables, such as the genotype of the coffee tree and its growing conditions (defined by the soil, climate, altitude, and other factors), post-harvest processing methods, the degree of the green coffee beans roasting, storage and distribution conditions, as well as coffee brewing methods [13,20,21,24,44,47,60,61,62]. Many studies have already been published in which elements [63,64,65,66,67,68,69,70,71], triglycerides [72], tocopherols [62,72,73,74], trigonellines [18,75,76,77,78,79], diterpenes [80,81,82], carbohydrates [76,83,84,85,86,87], polyphenols [36,37,39,65,75,76,77,78,88,89,90,91,92,93,94,95], 5-HMF [18,55,56], biogenic amines [79], amino acids [13,76,96], fatty acids [65,75,97,98,99], volatile compounds [10], and methylxanthines, especially caffeine [18,76,77,78,79,88,90,94,95], have been determined by various analytical methods to verify the coffee authenticity and quality.

## 2. Coffee Adulteration

According to the European Commission, food products are adulterated if their composition and/or quality do not match their description or labelling [100]. Adulterated food products are usually not harmful to health (sometimes the nutritional value may even be increased), but consumers have a right to know what exactly they are buying and consuming. Moreover, a potential risk of food allergies caused by additives has to be considered [15].

Coffee has been adulterated since time immemorial and today even ranks high at the top of the list of the most adulterated foods [101]. A very common method of coffee adulteration is to mix beans of different economic value. The undeclared, and thus illegal, addition of cheaper Robusta to Arabica is therefore considered a fraud. Therefore, many researchers have successfully studied the distinction between Arabica and Robusta in coffee blends [1,5,10,12,62,65,72,73,76,77,79,80,90,96,97,98,99,102,103,104,105]. As coffee quality is linked to specific growing areas, incorrect geographical indications are also considered illegal and have been verified by several teams of scientists [10,63,65,66,70,75,91,92,94,106,107,108]. The last common way of coffee adulteration is the blending of roasted coffee with undeclared materials. The list of coffee adulterants is very long and includes roasted and unroasted coffee husks and stalks, cereals (e.g., chicory, corn, barley, wheat, rye, oats, rice, buckwheat, triticale, bran, and malt), legumes (e.g., soybeans, peas, chickpeas, and carob), roots (e.g., chicory or dandelion), vegetables (e.g., potatoes, carrots, and beetroots), fruits (e.g., figs, bananas, acai, and prunes), nuts (e.g., almonds, peanuts, and chestnuts), and seeds (especially cocoa and sunflower seeds). Several techniques have also been developed to detect these impurities [10,15,83,84,85,86,87,107,109,110,111,112,113,114].

As indicated, adulteration practices are diverse and include many tricks to reduce production costs and thus increase the profit from the final product [112,115,116]. However, adulterated coffee products not only mislead consumers, but can also affect their health [84,86]. Therefore, it is essential that analytical techniques are able to detect various forms of adulteration (the use of poor-quality coffee beans, such as unripe, burnt, defective, etc. [10], presence of specific adulterants (Figure 2), degree of dilution, and unauthorized use of geographic origin of coffee beans [112,115,116]) to find whether the product label claims are based on the truth. For these reasons, various spectroscopic [12,63,64,67,68,69,70,71,75,80,81,82,106,107,108,113,117,118,119,120], electrophoretic [87,109], electrochemical [110,114], and biological [102,109,111,121] techniques have already been developed, but chromatographic techniques, especially high performance liquid chromatography (HPLC), have become the most dominant [1,47,61,72,73,85,86,88,90,94,122,123,124] for their more versatile use, reliability, reproducibility of results, possibility of automatization, identification of a large number of qualitative biomarkers in complex matrices, and low sample consumption [13,21,61,123,125].

## 3. By-Products of Coffee Industry

It has already been discussed and emphasized why it is crucial to examine coffee beans or coffee brews. However, the coffee business produces a huge amount of waste daily, which is contrary to the global trend of sustainability these days. Therefore, the coffee industry’s by-products are currently the subject of extensive research, particularly for potential future applications that could reduce the industry’s harmful effects on the environment [32,126]. As a result, numerous studies investigating the chemical composition and potential applications of the coffee waste, such as spent coffee grounds [22,121,127,128,129,130,131], coffee pulp [132,133], coffee silver-skin [40,129,134], and coffee leaves [135] (Figure 3), have been performed. The studies showed that by-products from the coffee industry still contain a significant amount of valuable ingredients, particularly caffeine and phenolic compounds, which are typically extracted and used as an additive in the dietary (sports supplements, functional foods, and food additives), cosmetic, and pharmaceutical industries.

## 4. Analysis of Antioxidants in Coffee Products Using HPLC

Determining the quality of coffee is still a very current issue, as evidenced by several reviews published on this topic in recent years. However, the vast majority of them deal with a summary of different analytical approaches used to identify and quantify various analytes, ultimately leading to the revelation of illicit and unauthorized practices [7,10,13,15,21,51,59,61,112,116,122,136]. Of the wide range of analytes present in coffee, this review is exclusively focused on the determination of the most important antioxidants, namely phenolic compounds (PPs) and caffeine, which are generally the most valued, analysed, and discussed because their content and profile can be used for coffee quality assessment [11,34,88,137,138,139]. HPLC is the most common technique for the qualitative and quantitative analysis of these compounds. Thus, the following overview summarizes the results of the latest strategies that have been developed and applied only for HPLC analysis of these analytes over the last 10 years, and contrasts the targeted analysis with the increasingly used modern method of non-targeted analysis, which seems to be cheaper, faster, and very effective (Figure 4).

### 4.1. Sample Preparation

Prior to chromatographic analysis, the samples of green, as well as roasted coffee beans, have to be always grinded to a powder from which the target analytes can be subsequently extracted. The contact surface, particle size, weight of the sieve, extraction technique used, as well as time, temperature, and pressure of extraction, are the most significant extraction variables [59]. Depending on the analyte, various extraction methods have already been developed. A simple solid–liquid infusion using hot water [11,34,47,59,88,139,140,141,142,143] or organic solvents [40,78,124,144] is the most common technique of PPs and caffeine isolation. Other extraction techniques, namely percolation [145], ultrasound-assisted [78,138,146] or microwave-assisted extractions [138], QuEChERS [78], and deep eutectic solvent-based extraction [95] have been used rarely. Since the coffee brewing method plays an essential role in the composition and health properties of the resulting beverage, many studies dealing with this issue have already been published [11,142,147,148,149]. In the study of Budryn et al. [147], the influence of coffee genotypes (Arabica vs. Robusta) on the efficiency of various extraction methods (brewing with boiling water and boiling in water at normal and elevated pressure) was investigated. The most efficient extraction methods of chlorogenic acid isomers from ground Arabica and Robusta green coffee beans were boiling with water at normal and elevated pressure, respectively. Since filter coffee is one of the most widely used coffee brew methods, and espresso coffee (a coffee beverage prepared by the pressure method) is the most appreciated by consumers, Ludwig et al. [148] compared these two preparation methods. It has been shown that in espresso preparation more than 70% of the antioxidants, especially chlorogenic acid isomers, were extracted from the coffee powder within the first eight seconds, whereas in filter coffee preparation, extraction started after 75 s. In the study by Rothwell et al. [149], the chemical profiles of 76 brewed coffee samples representing not only various brewing methods (classical instant coffee, espresso, K-cup pods, Turkish and Greek boiled coffees, drip machine, French Press, percolator, and cold brewing methods), but also different bean genotypes (Arabica and Arabica/Robusta blend), roasting levels (light, medium, and dark), and decaffeinated versions, were investigated using HPLC coupled with mass spectrometry (HPLC-MS), followed by principal component analysis (PCA). It was proven that the composition of coffee is strongly affected by all the variables mentioned above, with brewing methods being the main sources of chemical variability. Filtration and moka methods were compared in the study by Bobková et al. [11] in terms of contents of chlorogenic acid isomers and caffeine. In this study, all samples were Arabica beans roasted at medium-dark level. It was found that there is a significant difference in caffeine content between the analysed samples, while the change in chlorogenic acids content is not statistically significant. In the study by Miłek et al. [142], the influence of different brewing methods (aeropress, mocca i.e., percolator, and dripper) on the antioxidant capacity and caffeine content of the resulting infusions was also tested for two selected high-quality coffees. It was found that the use of a dripper provides the beverage with the best antioxidant properties, but a low concentration of caffeine.

### 4.2. Targeted Analysis

HPLC with spectrophotometric detection (HPLC-UV/VIS), combined with multivariate data treatment, was used to distinguish between specialty and traditional coffee beans in the study by Alcantara et al. [47]. Using PCA, all seventeen samples were very successfully divided into two groups (special versus traditional coffee) according to the quantity of caffeine, chlorogenic, nicotinic, and caffeic acids. This recognition of samples is useful for consumer protection because traditional coffees are of lower quality and, therefore, these can be purchased more cheaply. The main difference was the number and amounts of compounds that were responsible for the organoleptic properties of coffee. Traditional coffees, usually representing blends of Arabica and Robusta, contained higher caffeine and lower polyphenol contents than specialty coffees, which are typically composed of entirely 100% Arabica and roasted to a lower degree, resulting in less degradation of biologically active substances and, consequently, less loss of sensory properties.

A comparison of specialty coffee types of different botanical and geographical origin (seven Arabica samples and one Robusta sample) with commercial coffee brands (two samples) in terms of caffeine concentration, antioxidant capacity (determined by spectrophotometric DPPH and FRAP methods), and total phenolic content (determined by Folin-Ciocalteu method) was also performed in the study by Miłek et al. [142]. In addition to HPLC-UV/VIS, a reference spectrophotometric method was used for the determination of caffeine, which is based on the isolation of caffeine from alkalized infusion by liquid–liquid extraction into chloroform. The spectrophotometric results were lower than those obtained by HPLC due to the incomplete caffeine extraction into chloroform. The caffeine content of specialty coffee was similar to the commercial ones. On the other hand, the antioxidant capacity was significantly higher in specialty coffees. Regarding the differences between the specialty grade samples, Arabica coffee provided a great variability of caffeine, but its concentration was always lower than that of Robusta coffee. Further, the impact of the brewing method (aeropress, mocha, and dripper) on antioxidant activity and caffeine content in the final brews was tested for two selected high-quality coffees. It was found that the dripper resulted in a drink with the best antioxidant properties and moderate caffeine level.

The aim of the study by Muchtaridi et al. [34] was to determine the levels of caffeine and CafQA isomers in coffee beans from three different areas of West Java before and after their decaffeination. Decaffeination was performed by extracting the coffee powder with dichloromethane, followed by solid phase extraction into methanol. Then, the decaffeinated samples were analysed using HPLC-UV/VIS and also subjected to a neuraminidase binding assay to determine their biological activity. Caffeine and CafQA levels were found to affect neuraminidase inhibitory activity, and thus there is a correlation between these parameters.

The efficiency of the decaffeination process and its effect on the content of the three most abundant CafQA isomers was also studied by Klikarová et al. [88]. Authors proved that dichloromethane was the best extraction solvent, resulting in the most effective caffeine isolation with the least loss of CafQA isomers. Further, the impact of the roasting process was also assessed, and a large set of samples (64 regular (un)roasted and decaffeinated (un)roasted coffee beans in total) was subjected to the analysis. It was found that the caffeine concentration was almost not affected by the roasting process, whereas the significant loss of 5-CafQA (65–81%) was observed. The concentration of the other two chlorogenic acid isomers did not change considerably during the coffee beans’ roasting, even when they were roasted to the dark level. It is worth mentioning that this was the first time that HPLC-UV/VIS analysis of these target analytes has been carried out simultaneously, even in only a six-minute isocratic separation in which minimal unit peak resolution was attained.

The determination of the origin of American, African, and Asian coffee beans based on the chemical properties of the resulting beverage was performed in a study by Demianová et al. [139]. In fifteen samples of green coffee (five from America, five from Asia, and five from Africa), which were subsequently roasted to a medium level, the total antioxidant capacity (TAC) was determined spectrophotometrically using the DPPH method, and the contents of CafQA isomers and caffeine determined by the HPLC-UV/VIS method were assessed. For green coffee samples, the highest values of TAC and caffeine were found in American samples, while the highest content of CafQA isomers was observed in African samples. For roasted coffee samples, the values of TAC and CafQA isomers decreased by an average of 13.5% and 90%, respectively. On the contrary, the amount of caffeine increased by an average of 21.5%. A similar loss of 5-CafQA was observed in a study by Klikarová et al. [88]. However, no change in caffeine content was observed during roasting. Analysis of variance (ANOVA) single factor showed significant differences between green as well as roasted samples of different geographical origin in terms of the TAC and caffeine content [139]. However, CafQA content showed no difference.

The assessment and the comparison of antioxidant potential and content of selected biologically active substances (caffeine and coumaric, ferulic, caffeic, and chlorogenic acids) between green coffee samples and food supplements, based on green coffee extracts (Figure 5), was the aim of the work of Brzezicha et al. [146]. In fact, food supplements are not subject to any quality control or analytical verification of their composition before they are placed on the market. There are not even uniform procedures for verifying their authenticity. For this reason, many questions arise about their quality, efficacy, and safety, as well as whether a supplement or a food is a better source of biologically active substances. Brzezicha et al. [146] ascertained that green coffee samples have comparable or even higher antioxidant properties than dietary supplements. The amount of 5-CafQA in food supplements was very diverse (0.33–329 mg/g) compared to green coffee samples (32.7–47.6 mg/g). Moreover, the green coffee samples contained lower levels of caffeine. The authors found discrepancies between the determined and the manufacturer’s declared values (the amount of chlorogenic acid was in all cases below the declared value and, conversely, the amount of caffeine was higher in some samples than indicated on the packaging). The quality of food supplements could therefore be summarized as unsatisfactory.

Using HPLC, Cheserek et al. [137] characterized caffeine, chlorogenic acids, and other biochemical compounds in twenty samples of green coffee beans from Kenya, including different genotypes of Arabica, Robusta, and their variously crossed hybrids (Arabusta). The results were processed by PCA, and a correlation of chlorogenic acid with caffeine was noted. Robusta contained higher levels of caffeine and chlorogenic acids compared to Arabusta hybrids, which were more similar in composition to Arabica. The results showed that Arabusta hybrids surpassed Robusta coffee in the content and representation of biochemical compounds, which means successful introgression of quality genes.

A similar work was performed by Gutiérrez Ortiz et al. [143], who chromatographically determined all three pCoQA isomers and the sum of chlorogenic acids (CafQA, diCafQA, FQA, and pCoQA isomers) in 14 commercially available samples of *C. arabica*, *C. canephora*, and *C. liberica*, and 13 wild species of the genus Coffea (all samples were of different geographical origin coming from Honduras, Ceylon, French Guiana, East Afrika, France, Mozambique, Portugal, Brazil, Colombia, Ethiopia, India, Yemen, Vietnam, and Guatemala). It was the first time the distribution of all pCoQA isomers in wild-type coffee samples was described. The highest and the lowest content of pCoQA was obtained in the wild C. sessilifora (2.18 mg/g) and C. pseudozanguebariae (0.12 mg/g), respectively. The average pCoQA content in commercial green coffee beans was 0.55 mg/g. Furthermore, the influences of the roasting process and the geographical origin of commercial species on the distribution of pCoQA isomers, was evaluated.

The HPLC–MS instrumentation was used in a metabolomic approach analysing 76 coffee brew samples obtained by different brewing methods, roasting levels, bean species, and coffee types [149]. The study aimed to identify which characteristics of coffee most affect its chemical profile. The PCA statistical analysis (Figure 6) divided the samples according to the brewing method (instant coffee, espresso, and K-cup coffee brews). The clear discrimination between 100% Arabica samples and Arabica/Robusta blends was also observed in the same scatterplot (Figure 6). The high concentrations of six phenolic acid esters together with cafamarine and, simultaneously, the low concentrations of five diketopiperazines were the main descriptors of PC1, while PC2 was described by two feruloylquinic acid isomers, two phenolic acid amides, and five diketopiperazines. The differentiation between regular and decaffeinated coffee brews was achieved along PC3, which was explained by high loadings of paraxanthine and theobromine contents.

The content of caffeine, trigonelline, N-methylpyridinium, niacin, and chlorogenic acids present in 65 Italian capsules of caffeinated, as well as decaffeinated, coffee samples was examined using HPLC coupled to tandem mass spectrometry (HPLC-MS/MS). PCA showed wide variability both among capsules of the same brand and among different brands, which means that the content of bioactive compounds in a cup of coffee may vary significantly [140].

As decaffeinated green coffee bean extracts have beneficial effects and are sold as nutraceuticals or as dietary supplements, HPLC-MS was used to identify and quantify PPs in green, roasted, and spent coffee, as well as in coffee silverskin for the subsequent comparison of their levels in these products [40]. Silverskin extract has an overlapping PPs profile compared to green coffee bean extract. Moreover, this profile did not change even after silverskin decaffeination. Because caffeine is not always compatible with the nutraceutical purposes of CGA-containing extracts, decaffeinated silverskin can be considered a better raw material to produce CGA extracts than more expensive green coffee beans. This finding also contributes to the more sustainable impact of the coffee industry.

The influence of different degrees of roasting (green/unroasted, light, medium, and dark) on the phytochemical composition of Arabica beans was investigated by Montenegro et al. [138]. It is known that the roasting process affects the content of phytochemicals, and undesirable compounds may be formed. Thus, the amounts of caffeine, chlorogenic acids, and trigonelline present in coffee beans roasted to various degrees were analysed using HPLC-UV/VIS. Further, DPPH, ABTS, FRAP, and ORAC methods used to determine the antioxidant capacities of the extracts, as well as total phenolic content, were assessed. Subsequently, the presumed preventive effects of the consumption of coffee on the development of prostate cancer were evaluated. The target analytes were isolated from the samples by microwave extraction, which is an alternative to conventional extraction techniques because it preserves more bioactive compounds due to lower temperature and shorter time used. Extracts of green and light roasted coffee samples showed the highest antioxidant capacity, and thus, compared to medium and dark roasted coffee extracts, promoted higher inhibition of cell viability, caused greater cell cycle arrest, and more induced apoptosis. The caffeine content was not affected by roasting, while chlorogenic acid was degraded due to high temperature, and therefore its amount was lower in medium and dark roasted beans. This study demonstrated that the consumption of green and light roasted coffee extracts contributes to the inhibition of prostate tumour progression.

The determination of selected physico-chemical properties of coffee brew prepared by two different methods (filtration and moka method) from the same beans of Coffee Arabica roasted to a medium dark degree was investigated by Bobková et al. [11]. They focused on the analysis of dry matter, pH, and content of chlorogenic acids and caffeine, and the obtained data were evaluated by PCA and ANOVA. The caffeine content in the cup is highly dependent on the method of preparation, including the type of contact between coffee grounds and the solvent, roasting degree, extraction time, coffee/water ratio, temperature, the vapour pressure/boiling process, and caffeine kinetics [150]. In terms of caffeine content, ANOVA analysis revealed a significant difference between the two preparation procedures (found concentrations were between 1.37–1.78%), with samples prepared by the filtration method having lower caffeine content. CGA concentrations in filtered and moka coffee determined using HPLC-UV/VIS ranged between 1.41–2.94 g/100 g and 1.49–3.36 g/100 g, respectively. ANOVA found that these differences are statistically insignificant and, therefore, the CGA content is almost independent of the preparation method. These conclusions were further confirmed by PCA analysis [11].

The effect of the coffee roasting process on selected compounds was investigated by Macheiner et al. [144] and Schouten et al. [141] using HPLC-UV/VIS and HPLC-MS/MS instrumentation, respectively. Macheiner et al. [144] examined changes of CafQA and diCafQA isomers present in Arabica and Robusta coffee samples during different degrees and temperatures of roasting, batch size, and roaster designs, while Schouten et al. [141] focused on changes in antioxidant capacity (FRAP, DPPH, and ABTS methods), total phenolic content (Folin-Ciocalteu method), weight loss, water activity, density, moisture, and colour, as well as concentration changes of acrylamide, trigonelline, and nicotinic and caffeic acids in Arabica and Robusta coffee samples roasted to five different roasting degrees (light, medium-light, medium, medium-dark, and dark). Regardless of the botanical origin of the sample, the antioxidant capacity was highest in the first two stages of coffee roasting. With a higher degree of roasting, the antioxidant capacity decreased, but because of the formation of other antioxidant molecules, such as free quinic acid, melanoidins, or other low molecular weight phenolic compounds, the decline was only moderate [141]. Analogous findings concerning isomerization and other compositional changes occurring during the roasting process were also reported in the study by Klikarová et al. [88]. Further, Schouten et al. [141] presented that the total CGA content was higher in green and light roasted samples. The most abundant CGAs were 5-CafQA (about 80%), followed by 3-CafQA and 3,5-diCafQA. The content of 3-CafQA was increased by light roasting, while 5-CafQA was reduced or stagnated. Decreases in 5-CafQA, 3-CafQA, and 3,5-diCafQA of about 90%, 70%, and 70%, respectively, were observed in dark roasted samples. No significant differences in antioxidant capacity were found between the Robusta and Arabica green samples. However, after roasting, Robusta samples showed considerably higher values, probably due to higher caffeine content. In contrast, total CGA and trigonelline levels were higher in Arabica samples [141]. According to Macheiner et al. [144], chlorogenic acid isomerization reactions were detected at comparable stages of the coffee roasting process, regardless of species, variety, batch size, or roaster design. Degradation of 3-CafQA and 4-CafQA due to isomerization reactions were slower and occurred later in Robusta beans than in Arabica beans. Concentrations of 3,4-diCafQA and 4,5-diCafQA remained almost unchanged until the first crack, while 3,5-diCafQA degraded very rapidly regardless of Coffea species, batch size, and roaster designs. Thereafter, the concentrations of all diCafQA isomers observed continued to decrease until the end of the roasting process.

Screening of five chlorogenic acids and caffeine in green coffee beans using a low-pressure liquid chromatography (on 1-cm length monolithic column) with amperometric detection was performed by Silva et al. [124]. Their method was rapid, low-cost, user-friendly, and generated low waste volumes. This instrumentation has proven to be an efficient and versatile technique capable of performing automatic sample processing at high speed. For the above mentioned reasons, it is considered to be competitive for the conventional HPLC method.

Chemical composition (phenolic compounds and caffeine content), selected physico-chemical properties, and antioxidant activity of 26 conventional and 19 organic coffee samples coming from the main Brazilian production regions were evaluated using various chemometric tools, such as PCA, linear discriminant analysis (LDA), partial least squares regression combined with discriminant analysis (PLS-DA), data-driven soft independent modelling of class analogy (DD-SIMCA), support vector machines (SVM), and k-nearest neighbors (k-NN) [60]. Organic and conventional coffee samples have different cultivation system and were successfully distinguished using PCA. However, their distinction among the production regions or botanical origin was not achieved by this statistical method. On the other hand, PLS-DA, LDA, SVM, and k-NN could discriminate practically all samples based on both cultivation systems and coffee varieties. The monitored parameters did not significantly depend on the geographical origin of the coffee, which could therefore not be estimated.

The most typical parameters examined in the HPLC targeted analysis (cultivation of the coffee plant, origin of the coffee beans, and their subsequent processing), as well as the corresponding preparation of the sample for analysis, the detection used, and any statistical data processing, are compiled in Table 1.

### 4.3. Non-Targeted Analysis

Recently, numerous non-targeted analysis approaches have been developed, dealing not only with the HPLC fingerprints [1,49,151,152,153,154,155,156,157,158,159,160], but also, less frequently, with profiling using techniques such as gas chromatography coupled to mass spectrometry [19,125,161], nuclear magnetic resonance (NMR) [103,115], UV/VIS spectroscopy [162], or inductively coupled plasma optical emission spectrometry [66,70]. These techniques are predominantly combined with multidimensional statistical methods, such as PCA, factor analysis (FA), discriminant analysis (DA), partial least squares regression (PLS), and their combinations (e.g., PLS-DA), in order to obtain as much information from the measured data as possible.

Strategies of non-targeted chromatographic fingerprinting are based on recording instrumental signals as a function of retention time, but without knowing any further information (identification or quantification) about the compounds providing these signals. For this purpose, simple sample processing procedures are usually used to obtain as many compounds of different families as possible [151]. Thus, non-targeted analysis represents a very simple, rapid, and inexpensive method that could be advantageously used to verify the authenticity and quality of coffee.

Non-targeted HPLC-MS metabolic profiling was effectively used to elucidate the relationship between metabolites and the cupping score indicating the beverage quality [152]. In total, thirty-six varieties of green beans from Guatemala were subjected to the analysis. Using an orthogonal partial least squares (OPLS) regression model, two metabolites (from a total of 2649 valid peaks) were found to be strongly correlated with a high cupping score, and can therefore be utilised as universal quality indicators. The metabolites were first purified and then spectroscopically identified as isomers of 3-methylbutanoyl disaccharides (i.e., precursors of 3-methylbutanoic acid that is known to enhance the coffee quality).

Similar methodology was presented in the study by Sittipod et al. [153], who employed non-targeted HPLC-MS profiling of eighteen coffee samples, together with OPLS analysis, to find chemicals that enhance the coffee flavour quality. Despite the fact that four compounds positively correlated with the cup score were isolated and purified, only three of them were confirmed by sensory recombination analysis (performed by certified Specialty Coffee Association Q-graders) as indicators that significantly increased the cup scores. Subsequently, using NMR and high-resolution MS, these compounds were identified as novel derivatives of 3-methylbutanoylquinic acid. Although none of them showed any direct flavour activity, it can be argued that they act as flavour modifiers.

Another approach to non-targeted metabolomic analysis, based on ultra HPLC-MS analysis combined with statistical processing of measured data, was presented by Xu et al. [154]. Using PCA and hierarchical clustering analysis, all samples were successfully divided into three clusters according to the brewing method (pour-over, boiled, and cold brew). Subsequently, the OPLS-DA model revealed nine potential markers, five of which (norharman, harman, pyrimethanil 1-palmitoyllysophosphatidylcholine, and 4-hydroxy-3-methoxycinnamaldehyde) were consequently confirmed by HPLC-MS using the reference standard, and these can be considered as characteristic brewing markers. Interestingly, the cold-brew samples were richer in harmane and norharmane contents than the heat-treated (boiled and pour-over) samples.

For characterisation and evaluation of the coffee authenticity and quality, a total of five papers concerning the non-targeted HPLC fingerprint strategies using UV/VIS or fluorescent detection (FLD), combined with chemometrics, were published by the Spanish researchers in 2020–2021 [1,151,155,156,157]. In 2020, they analysed a total of 306 commercially available coffee samples, of which 240 were Nespresso-type products of various origins (Nicaragua, Brazil, India, Uganda, Ethiopia, Central/South America, Columbia, or Indonesia), purchased in supermarkets in Barcelona (Spain), and brewed directly by using an espresso machine [1]. The next 66 samples were purchased in bean form in Vietnam and Cambodia and, after grinding, these were brewed using a moka pot coffee maker. All samples differed in variety (Arabica, Robusta, or their mixture) and degree of roasting (1–5). Selected samples were also used for adulteration studies where the original coffee was mixed with “adulterant” coffee (Colombia vs. Ethiopia, Colombia vs. Nicaragua, India vs. Indonesia, Vietnam-Arabica vs. Vietnam-Robusta, Vietnam-Arabica vs. Cambodia, and Vietnam-Robusta vs. Cambodia) in various ratios ranging between 100:0–0:100 (original coffee: adulterant coffee; *w*/*w*). HPLC-UV/VIS fingerprints (Figure 7) were subjected to statistical analysis (PCA, PLS-DA, and PLS regression) and found to be sufficient chemical descriptors to classify coffee by geographical origin (even for nearby countries such as Vietnam and Cambodia), varieties, and degree of roasting (Figure 8). Regarding botanical origin (variety), the differences are mainly based on the relative intensities of the peak signals, as the fingerprint profiles are similar (Figure 7). Additionally, PLS regression could reveal coffee adulteration down to 15% of adulterant coffee (coffee of a different geographical or botanical origin than declared) [1]. All 66 Vietnamese and Cambodian samples, together with half of the Nespresso-type samples, both processed as before, were also analysed by HPLC-FLD to obtain fingerprints that were consequently subjected to PCA and PLS-DA statistical analysis as well [155]. HPLC-FLD fingerprints of only two Vietnamese, one Cambodian, and five Nespresso-type coffee samples were again used to reveal adulteration cases related to different production regions. For this purpose, the same pairs of original coffee and adulterant coffee (Colombia vs. Ethiopia, Colombia vs. Nicaragua, India vs. Indonesia, Vietnam-Arabica vs. Vietnam-Robusta, Vietnam-Arabica vs. Cambodia, and Vietnam-Robusta vs. Cambodia) were compared [156]. From these two papers, the identical conclusions as in the previous work published in 2020 were interpreted.

Further, both HPLC-UV/VIS and HPLC-FLD fingerprints of only 54 previous samples of Vietnamese and Cambodian coffee, together with 69 samples of chicory, flour (wheat, rice, cornmeal, rye, and oatmeal), and barley, which were subsequently mixed into coffee as adulterant in ratios ranging between 100:0–0:100 (coffee:impurity; *w*/*w*), were evaluated using PLS-DA to determine the adulteration level [151]. Various extraction solvents (water, methanol, ethanol, acetonitrile, acetone, and organic-aqueous mixtures containing 20, 50, and 80% of each organic component examined) were tested to obtain the maximum number of signals. The highest extraction capacity was achieved by using H_2_O:acetonitrile (50:50, *v*/*v*) and H_2_O:methanol (50:50, *v*/*v*) for FLD and UV/VIS detection, respectively. Coffee adulterants provided completely different fingerprints than coffee samples, and their amount could be detected down to 15%. Comparing both fingerprint techniques (Figure 9), HPLC-FLD fingerprints did not completely distinguish coffee from barley samples, while all samples were perfectly discriminated by HPLC-UV/VIS fingerprints [151]. A comparison of the applicability of HPLC-UV/VIS and HPLC-FLD fingerprints for the detection and quantification of chicory present in instant regular (40 samples) and decaffeinated (26 samples) coffee was again performed by the same group of Spanish researchers [157]. In addition to coffee samples, 22 ground, as well as instant chicory samples, were analysed. Instant samples were prepared by their dissolving in hot water, while the ground samples were brewed using a moka pot coffee maker. Regarding the statistical analysis, PCA was an exploratory method employed to evaluate the performance of the quality control solution and ensure the robustness of chemometric data processing, PLS-DA was a classification method (regular coffee, decaffeinated coffee, and chicory), and PLS served as a multidimensional calibration method for the quantification of chicory in cases of determination of coffee adulteration level. Based on both HPLC fingerprints, samples were possibly distinguished into three groups according to their characteristics and coffee fraud was detected down to 15% of chicory content. Although HPLC-UV/VIS fingerprints were better to distinguish between regular and decaffeinated coffee, the HPLC-FLD method provided better linearity and error of calibration, as well as lower prediction errors.

A simple HPLC fingerprint method, together with simultaneous determination of selected bioactive compounds, was developed to evaluate the quality of twenty-four *C. arabica* samples of different geographical origin [158]. About 50 peaks were observed in the fingerprint. However, only thirteen intense peaks with good resolution characterizing the sample were selected. Correlation analysis and PCA analysis proved that the combination of HPLC fingerprint and quantitative analysis can be an effective tool for the evaluation of coffee quality.

A group of Brazilian scientists focused on the fingerprints of different cultivars of *C. arabica* L., namely traditional red Bourbon cultivar declared as a pure Arabica coffee, without breeding or crossing with other cultivars, and genetically modified hybrids IAPAR59, IPR101, and IPR108 that have been originated by crossing [159,160]. In both works, four organic solvents (ethanol, ethyl acetate, dichloromethane, and hexane) and their mixtures (15 attempts in total) were tested for ultrasonic extraction of substances present in these cultivars. The best extraction solvent was selected using a multivariate statistical design, followed by PCA data treatment [159], or parallel factor analysis (PARAFAC) [160]. According to PCA analysis of the HPLC-UV/VIS and infrared spectroscopy fingerprints (acquired after extraction with all 15 solvent mixtures), ethanol-dichloromethane (1:1) was the best extractant for distinguishing between the cultivars. The absorptions of HPLC-UV/VIS spectra recorded at 275 nm correlate with the intensities of the infrared absorptions between 3400–3460 cm^−1^, and can be explained by different levels of caffeine in the cultivars tested [159]. On the contrary, in Guizellini’s research [160], higher extraction efficiency was attained using the mixture of ethanol, dichloromethane, and hexane, and the mixture of all four solvents for the Bourbon and IPR101 cultivars, respectively. The three-way PARAFAC strategy determines the correlations of chromatographic and spectral data simultaneously, allowing a clearer assignment of metabolic groups than can be acquired by conventional HPLC-UV/VIS data treatment.

Not only the influence of variety (Arabica or Robusta), but especially the effect of roasting conditions on the near-infrared radiation (NIR) and HPLC-UV/VIS profiles of coffee, was investigated in the study by De Luca et al. [49]. The data were processed using ANOVA-simultaneous component analysis, which allowed the characterization of entire instrumental profiles, thus providing a holistic characterization of the roasting process and the authentication of individual coffee beans. By processing the NIR data, it was found that both the variety and the roasting time significantly affect the spectral profile. Moreover, PLS-DA and SIMCA were applied to NIR fingerprints data to verify the botanical origin of coffee beans. PLS-DA resulted in approximately 98% correct classification, and the sensitivity and specificity values were usually above 90% using SIMCA. Similar findings were obtained with chromatographic profiles. Almost all analytes detected by HPLC had lower concentrations in Arabica samples than in Robusta samples. As for the effect of roasting time, the intensity of almost all peaks decreased with increasing roasting time. 

The following overview (Table 2) compiles not only the conditions of non-targeted HPLC analysis (sample pre-treatment together with detection and statistical data processing used), but also unambiguously provides information on individual studies dealing with various factors affecting the coffee quality, such as the origin of coffee beans and their post-processing, as well as adulteration studies and examination of cupping scores.

## 5. Discussion and Conclusions

As coffee consumption rises every year, the demand and pressure on its very high quality also increases. For this reason, effective methodologies for analysing the chemical composition of coffee, and thus verifying its authenticity and quality, are still being sought.

Combining HPLC with chemometric methods has proven to be an indispensable tool in the discovery of descriptors capable of detecting differences between samples regarding product quality, geographical origin of production, genotypes, forms of cultivation (conventional or organic), roasting degree, brewing methods, etc. In this overview, we summarized the latest trends in the methods of targeted and non-targeted HPLC analysis of the most dominant coffee antioxidants used to not only confirm the authenticity of coffee, but also to reveal how the production process (roasting, storage, etc.), along with the coffee brewing methods, affect the composition of the resulting beverage.

Figure 10 clearly illustrates the key benefits and drawbacks of targeted and non-targeted analysis. Regarding targeted analysis, it provides very valuable information about the occurrence and concentration of selected (usually significant) analytes in the sample, even without statistical processing of the data obtained. Unfortunately, this qualitative and quantitative determination cannot be performed without the acquisition of frequently expensive analytical standards and the application of any quantitative method requiring additional analyses associated with increased consumption of chemicals. If we consider also the time-consuming development of an extraction method suitable for selected analytes (with high recovery), and the long optimization of HPLC separation, which must provide sufficiently separated peaks with good resolution, targeted analysis then represents a relatively time-, financially-, and manually demanding multi-step approach. Although technological progress has made it possible to detect fraudulent practices in coffee by determining specific chemical or biological markers with higher sensitivity than ever before, it can be argued that targeted analysis is unable to reveal all common counterfeiting practices, and thus its application is only limited in this field.

On the other hand, in non-targeted analysis (sample fingerprinting/profiling), the traditional procedure of determining analytes in the sample is skipped (Figure 4) because it is not crucial to know which analytes the sample contains, let alone in what quantity. This indicates that we do not need any analytical standards for the identification of given peaks, nor for their subsequent quantification by some quantitative method (e.g., calibration curve method, multiple standard addition method, method of direct comparison, etc.). In non-targeted analysis, even the optimization of extraction and separation differs from that one used in standard targeted analysis. In this case, the goal is simply to get as many peaks as possible and thus the richest possible chromatogram. Thanks to the easy and rapid optimization of sample pre-treatment and separation, no preparation of calibration solutions, and no identification and quantification of peaks, we significantly reduce the final costs and time. As a result, less demands are placed on the operator, which also reduces the rate of errors. The entire process of non-targeted analysis is complicated only by the final (but mandatory) step of statistical data treatment. Just employing the multivariate statistical methods leads to reliable revelation of samples that were falsified by various known practices. In conclusion, we can summarize that the studies dealing with non-targeted analysis are able to obtain a large amount of information (without the need for specific qualitative and quantitative analysis), making the entire analysis much faster and more informative, accurate, efficient, and mainly more suitable for sample quality assessment.

## Figures and Tables

**Figure 1 molecules-27-07419-f001:**
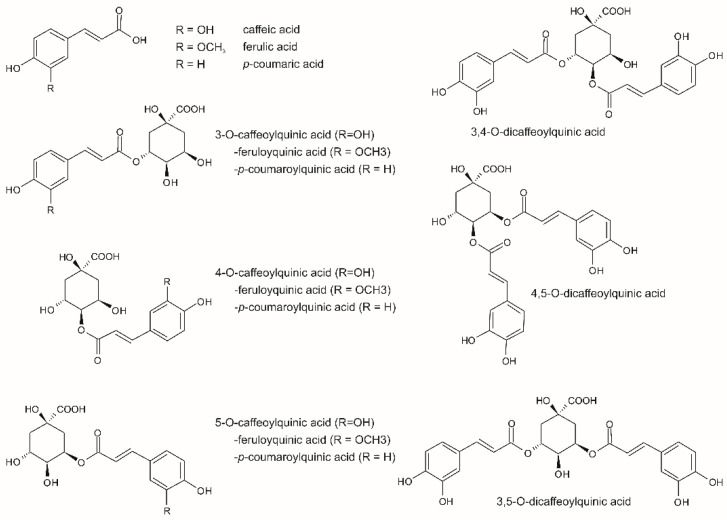
Structures of the most important chlorogenic acids present in coffee.

**Figure 2 molecules-27-07419-f002:**
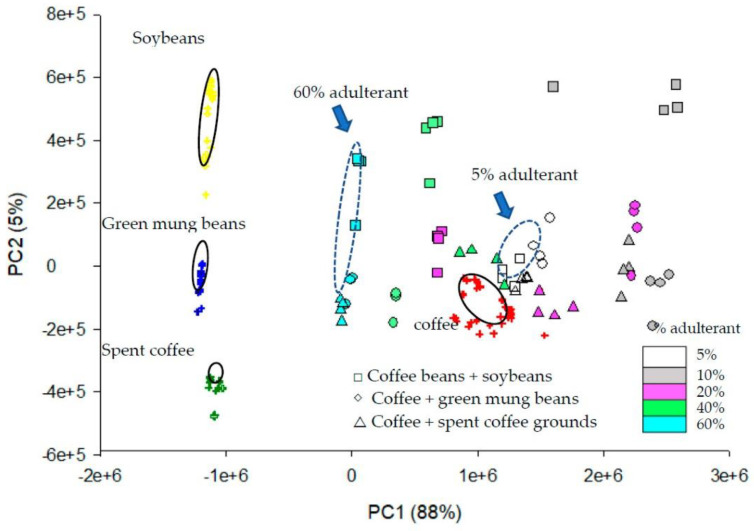
Distinguishing coffee beans from soybeans, green mung beans, spent coffee grounds and adulterated coffee containing different adulterants in different mixing ratios (5%, 10%, 20%, 40%, and 60%) using PCA [123].

**Figure 3 molecules-27-07419-f003:**
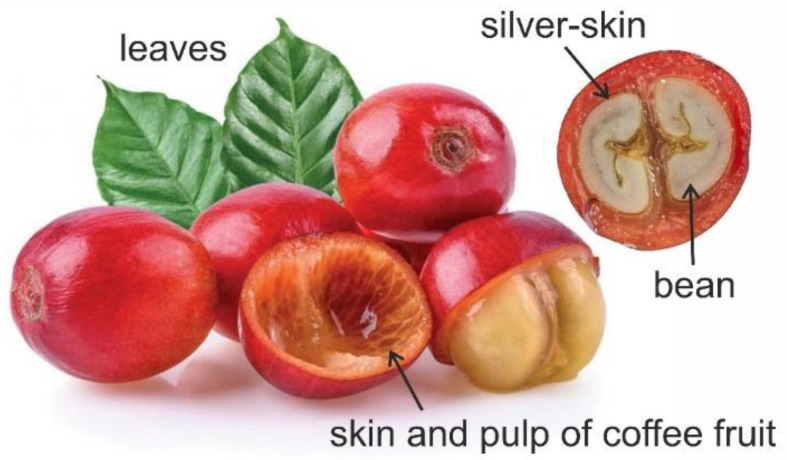
Coffee fruit anatomy.

**Figure 4 molecules-27-07419-f004:**
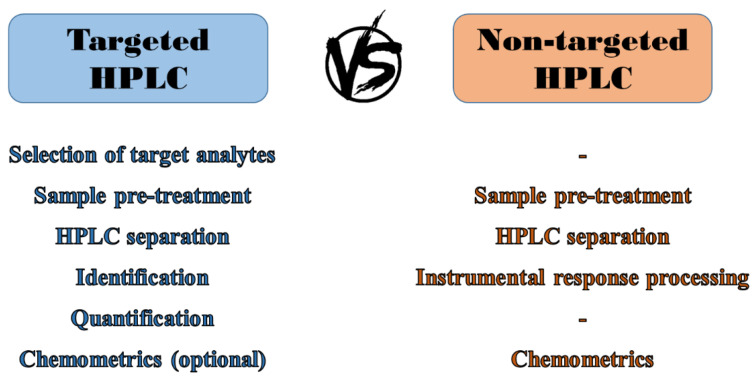
Basic diagram of the process of targeted and non-targeted analysis.

**Figure 5 molecules-27-07419-f005:**
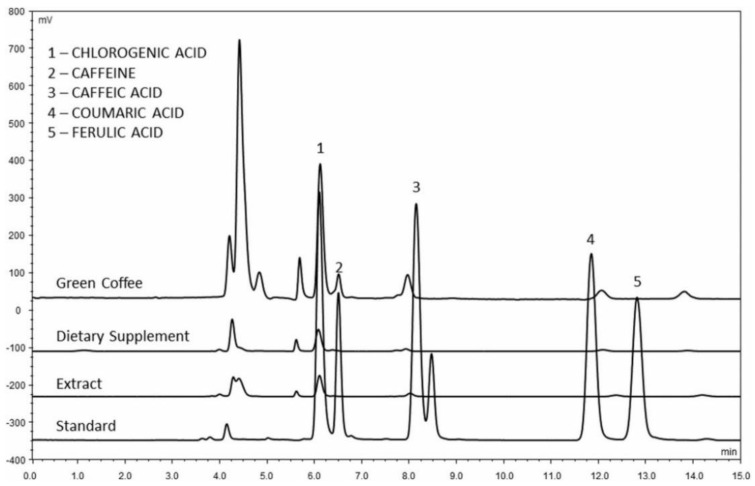
HPLC separation of chlorogenic acid (1), caffeine (2), caffeic acid (3), coumaric acid (4), and ferulic acid (5) present in ground green coffee, dietary supplement, and green coffee extract [146].

**Figure 6 molecules-27-07419-f006:**
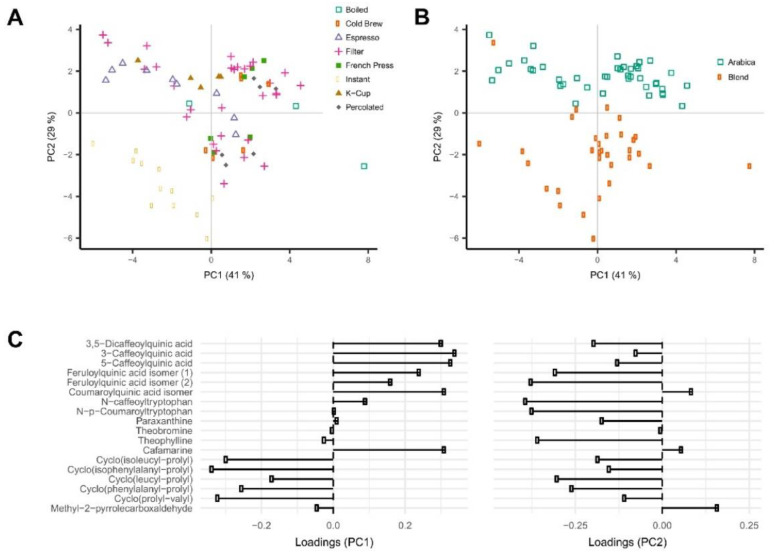
Discrimination of 76 coffee samples using principle component analysis: PCA scores of different coffee brews (**A**) and different coffee beans variety (**B**) and corresponding PCA loading plots (**C**) [149].

**Figure 7 molecules-27-07419-f007:**
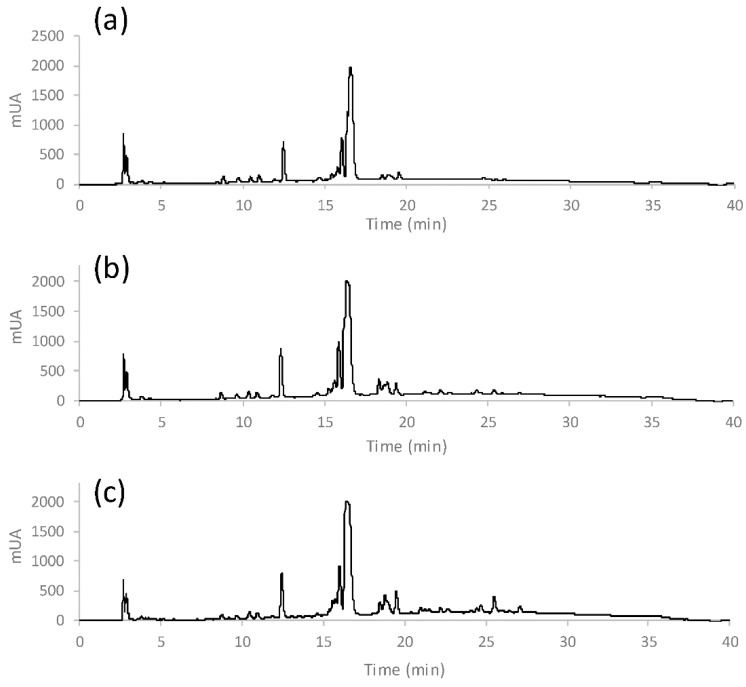
Non-targeted HPLC-UV/VIS fingerprints of Arabica coffee from Ethiopia (**a**), Arabica-Robusta mixture from India (**b**), and Robusta coffee from Uganda (**c**) [1].

**Figure 8 molecules-27-07419-f008:**
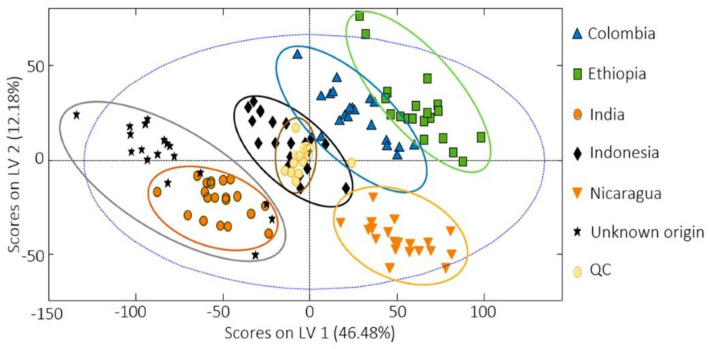
PLS-DA differentiation of coffee samples based on their geographical origin [1].

**Figure 9 molecules-27-07419-f009:**
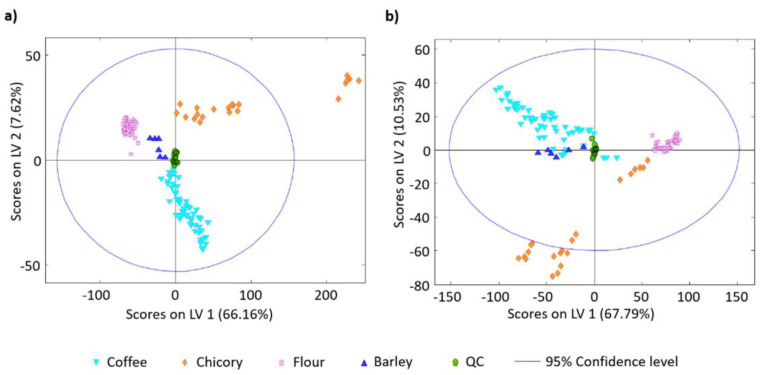
PLS-DA classification of the samples using HPLC-UV/VIS (**a**) and HPLC-FLD (**b**) fingerprints [151].

**Figure 10 molecules-27-07419-f010:**
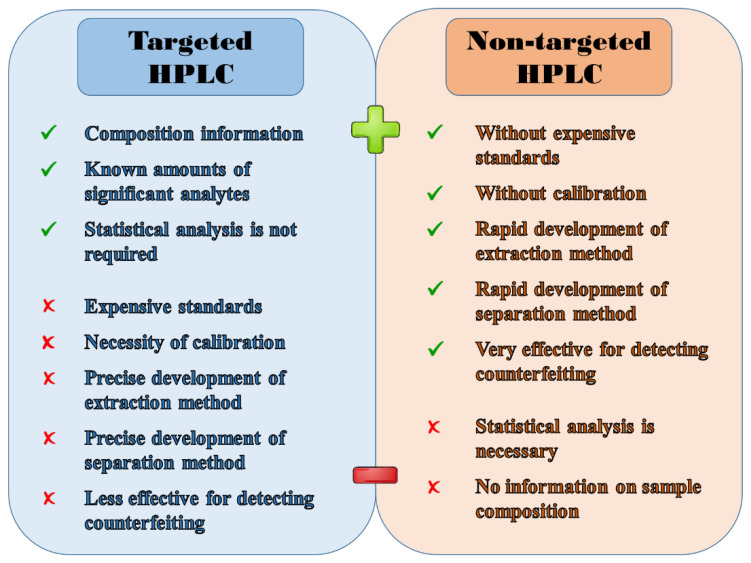
Principal benefits and drawbacks of targeted and non-targeted analysis.

**Table 1 molecules-27-07419-t001:** Overview of targeted HPLC analysis.

Kind of Sample	Sample Pre-Treatment	Impact of Cultivation, Brewing, Roasting, Decaffeination, Botanical and/or Geographical Origin	Detection	Statistical Data Treatment	Ref.
Arabica roasted coffee beans	s–l (hot water)	Br	UV/VIS	PCA and ANOVA	[11]
Coffee beans from West Java	s–l (hot water)	D	UV/VIS	-	[34]
Green, roasted, and spent coffee and coffee silverskin	s–l (organic solvents)	R and D	UV/VIS and MS	PCA	[40]
Specialty and traditional coffee beans	s–l (hot water)	-	UV/VIS	PCA	[47]
Conventional and organic Brazilian coffee samples	s–l (hot water)	C; B; and G	UV/VIS	DA, PLS-DA, DD-SIMCA, SVM, and k-NN	[60]
Roasted and green coffee beans	s–l (hot water)	R; D; B; and G	UV/VIS	PCA, FA, and ANOVA	[88]
Green coffee beans	s–l (organic solvents)	-	AMP	-	[124]
Kenyan green coffee beans and their hybrids	NS	B	NS	PCA	[137]
Arabica roasted and green coffee beans	UAE and MAE	R	UV/VIS	-	[138]
American, African, and Asian green coffee beans	s–l (hot water)	R and G	UV/VIS	ANOVA	[139]
Italian capsules of (de) caffeinated coffee	s–l (hot water)	D	MS/MS	PCA	[140]
Arabica and Robusta coffee beans	s–l (hot water)	R and B	MS/MS	ANOVA	[141]
Specialty and commercial coffee beans	s–l (hot water)	Br; B; and G	UV/VIS	-	[142]
Commercially available coffee beans and wild species	s–l (hot water)	R; B; and G	UV/VIS	-	[143]
Arabica and Robusta coffee beans	s–l (organic solvents)	R and B	UV/VIS	-	[144]
Green coffee beans and food supplements	UAE	-	Corona	-	[146]
Roasted coffee beans	depends on the brewing	Br; R; D; B; and G	MS	PCA	[149]

Abbreviations: AMP, amperometrical detection; ANOVA, analysis of variance; B, botanical origin; Br, brewing method; C, cultivation system; D, decaffeination; DA, discriminant analysis; DD-SIMCA, data-driven soft independent modelling of class analogy; G, geographical origin; k-NN, k-nearest neighbors; MAE, microwave-assisted extractions; MS, mass spectrometry; NS, not specified; PCA, principal component analysis; PLS, partial least squares regression; R, roasting; Ref., reference; s–l, solid–liquid infusion; SVM, support vector machines; UAE, ultrasound-assisted extraction; UV/VIS, ultraviolet/visible.

**Table 2 molecules-27-07419-t002:** Overview of non-targeted HPLC analysis.

Kind of Sample (Number)	Sample Pre-Treatment	Impact of Brewing, Roasting, Decaffeination, Botanical and/or Geographical Origin	Adulteration Studies, Cupping Score Investigation	Detection	Statistical Data Treatment	Ref.
Commercially available Arabica, Robusta, and their mixtures (306)	Nespresso machine or moka	R; B; and G	Adulteration	UV/VIS	PCA, PLS, and PLS-DA	[1]
Arabica and Robusta beans (6)	s–l (hot water)	R and B	-	UV/VIS (+IR fingerprint)	ASCA, PLS-DA, SIMCA	[49]
Vietnamese Arabica, Robusta coffee beans, their mixtures and samples of chicory, barley, and flour (123)	s–l (water-MeOH mixture)	B and G	Adulteration	UV/VIS and FLD	PLS and PLS-DA	[151]
Arabica beans (40)	s–l (hot water-MeOH mixture)	-	Cupping score	MS	OPLS	[152]
Arabica and Robusta beans (18)	drip-coffee maker and clean-up (SPE)	G	Cupping score	MS	OPLS	[153]
Arabica beans (3)	s–l (hot/cold water)	Br and G	-	MS	ANOVA, PCA, HCA, and OPLS-DA	[154]
Commercially available Arabica, Robusta, and their mixtures (186)	Nespresso machine or moka	R; B; and G	Adulteration	FLD	PCA and PLS-DA	[155]
Commercially available Arabica, Robusta, and their mixtures (8)	Nespresso machine or moka	R; B; and G	Adulteration	FLD	PLS and PLS-DA	[156]
Regular and decaffeinated coffee beans and instant and ground chicory samples (88)	moka or s–l (hot water)	D	Adulteration	UV/VIS and FLD	PCA and PLS-DA	[157]
Arabica coffee beans (24)	NS	G	-	NS	PCA	[158]
Green beans of different cultivars of Arabica (4)	UAE (organic solvents)	B	-	UV/VIS (+IR fingerprint)	PCA	[159]
Green beans of different cultivars of Arabica (2)	UAE (organic solvents)	B	-	UV/VIS	PARAFAC	[160]

Abbreviations: ANOVA, analysis of variance; ASCA, anova-simultaneous component analysis; B, botanical origin; Br, brewing method; D, decaffeination; DA, discriminant analysis; HCA, hierarchical clustering analysis; SIMCA, soft independent modelling of class analogy; G, geographical origin; IR, infrared; MeOH, methanol; MS, mass spectrometry; NS, not specified; OPLS, orthogonal partial least squares regression model; PCA, principal component analysis; PARAFAC, parallel factor analysis; PLS, partial least squares regression; R, roasting; Ref., reference; s–l, solid–liquid infusion; SPE, solid-phase extraction; UAE, ultrasound-assisted extraction; UV/VIS, ultraviolet/visible.

## Data Availability

Not applicable.

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
