# Peer review of "Targeted and Non-Targeted HPLC Analysis of Coffee-Based Products as Effective Tools for Evaluating the Coffee Authenticity"

_molecules, 2022, doi:10.3390/molecules27217419_

Round 1

Reviewer 1 Report

This is a very interesting review manuscript addressing an important adulteration topic: coffee fraudulent practices, which are increasing in the last  years, being among the top 10 food and beverage products more susceptible of fraudulent practices. In general the review is well written, and both targeted and non-targeted HPLC strategies have been addressed. In my opinion the manuscript is quite interesting for the scientific community dealing with cofffee authentication issues, but also in any food authentication problem as the described methodologies are general strategies in the food authentication field.

The authors are carrying out a througouht description of many of the reported applications dealing with targeted and non-targeted HPLC strategies for coffee characterization and authentication, in some cases in much detail. However, in my opinion, the review manuscript lacks a critical evaluation of these methodologies by comparing them and highlighting the advantages and disadvantages of each commented application. For example, many of the described applications are emplying chemometrics to address classification issues, but after reading the review manuscript is difficult to see which kind of methodology, targeted or non-targeted (if any of them) is providing the best results in terms of sample classification capabilities, etc. So I recommend the authors to try to compare, at some level, the reported data in a more critical way, not only from a merely descriptive point of view. Advantages and disadvantages of bot non-targeted and targeted strategies based on the data reported in the literature on coffee authentication is required.

Besides, I also suggest to illustrate some of the applications with figures, that allow to better understand and visualize what the authors are commenting (authors can just ask permission to reproduce the figure data on the review mansucript). 

Minor thinds:

- Tittle shouls be corrected: "... as an effective tool..." or "...as effective tools..."

- Figure 1 quality need to be improved.

- Line 246: At a point after reference [142]

Author Response

Reviewer 1:

This is a very interesting review manuscript addressing an important adulteration topic: coffee fraudulent practices, which are increasing in the last years, being among the top 10 food and beverage products more susceptible of fraudulent practices. In general the review is well written, and both targeted and non-targeted HPLC strategies have been addressed. In my opinion the manuscript is quite interesting for the scientific community dealing with cofffee authentication issues, but also in any food authentication problem as the described methodologies are general strategies in the food authentication field. The authors are carrying out a througouht description of many of the reported applications dealing with targeted and non-targeted HPLC strategies for coffee characterization and authentication, in some cases in much detail. However, in my opinion, the review manuscript lacks a critical evaluation of these methodologies by comparing them and highlighting the advantages and disadvantages of each commented application. For example, many of the described applications are emplying chemometrics to address classification issues, but after reading the review manuscript is difficult to see which kind of methodology, targeted or non-targeted (if any of them) is providing the best results in terms of sample classification capabilities, etc. So I recommend the authors to try to compare, at some level, the reported data in a more critical way, not only from a merely descriptive point of view. Advantages and disadvantages of bot non-targeted and targeted strategies based on the data reported in the literature on coffee authentication is required.

Thank you very much for your valuable comments and suggestions. The last chapter has been renamed “Discussion and conclusion”, and it includes additional discussion of the differences between targeted and non-targeted analysis, thus the critical evaluation of these two methodologies is now implied in the manuscript. Moreover, we have added the descriptive figure with advantages and disadvantages of both methodologies. We hope you find it satisfactory.

Besides, I also suggest to illustrate some of the applications with figures, that allow to better understand and visualize what the authors are commenting (authors can just ask permission to reproduce the figure data on the review manuscript).

Thank you very much for your suggestion. We have added eight figures and two tables to the revised manuscript. We hope that a total of ten figures and two tables is sufficient amount.

Minor thinks:

- Tittle should be corrected: "... as an effective tool..." or "...as effective tools..."

Thank you very much for the language correction. It was corrected accordingly.

- Figure 1 quality need to be improved.

Thank you for the warning. The resolution and format of the figure was changed (600DPI, TIFF).

- Line 246: At a point after reference [142]

Thank you for correction. It was change accordingly.

Reviewer 2 Report

The introduction on the history and production of coffee attracts attention. Collecting and reviewing 162 references, the authors made huge efforts to summarize the most important results in the field of targeted and nontargeted HPLC analysis of coffee-based products.

The main problem on the manuscript is, that the topic belongs to a journal in the field of food chemistry or food research. The manuscript is focusing mainly on various HPLC methods, thus a chromatographic journal would be also a suitable place for publication of this manuscript.

A few general remarks:

The numbering of references in the text should follow each other in increasing order. The abbreviation should be always explained, where it occurs at first. The conclusion is too short and meaningless.

Author Response

Reviewer 2:

The introduction on the history and production of coffee attracts attention. Collecting and reviewing 162 references, the authors made huge efforts to summarize the most important results in the field of targeted and nontargeted HPLC analysis of coffee-based products.

The main problem on the manuscript is, that the topic belongs to a journal in the field of food chemistry or food research. The manuscript is focusing mainly on various HPLC methods, thus a chromatographic journal would be also a suitable place for publication of this manuscript.

Thank you for evaluating our manuscript. You are right, the topic belongs to a food science journal, however, we received an invitation to this special issue of Molecules, which is dedicated to biologically active substances. We discussed with editors about the possibility of publishing our work in this issue and they were open to this idea.

A few general remarks:

The numbering of references in the text should follow each other in increasing order. The abbreviation should be always explained, where it occurs at first. The conclusion is too short and meaningless.

Thank you for your comments. The increasing order of references was followed, but some of them were used more than once, therefore, lower reference numbers also appear later in the text, especially in the chapters related to targeted and non-targeted analyses, which are written according to a logical key and not in order of increasing reference number (which has already been mentioned earlier in the manuscript). Nevertheless, we managed to renumber some references in the non-targeted analysis chapter, where we tried to keep the increasing numbers of citations.

The explanation of abbreviations has been reviewed and corrected. Thank you for your remark. Only the abbreviations of spectrophotometric methods used for antioxidant capacity assessment (e.g. DPPH, FRAP, etc.) were not explained, because these are established commonly used abbreviations. We even considered removing these detailed specifications from the manuscript and leaving only the information about the antioxidant capacity measurement. However, we believe that information about the specific methods used could be interesting for readers.

Finally, the conclusion was extended and renamed “Discussion and conclusion”.

We hope you find all the changes satisfactory for publishing our manuscript.

Reviewer 3 Report

The subject of the article arouses interest, but the summary leaves something to be desired. It should reflect more clearly the content of the article.

The introduction is a bit long but with good quality and well-written information. 

The following parts of the article are hard to follow but the information very valuable. I suggest, if possible, a better organization of the material to make it easy to follow

Author Response

Reviewer 3:

The subject of the article arouses interest, but the summary leaves something to be desired. It should reflect more clearly the content of the article. The introduction is a bit long but with good quality and well-written information. The following parts of the article are hard to follow but the information very valuable. I suggest, if possible, a better organization of the material to make it easy to follow

Thank you very much for your comment. We have added a couple of figures and tables to make the text more interesting, readable, and less ponderous for the readers, thus easier to follow.

Reviewer 4 Report

Dear Authors and editors,

this is a comprehensive review of the targeted and non-targeted analysis of coffee and its products using LC and MS. The review is detailed enough, English language is OK as far as I can judge, the number of cited references is adequate for a review article and reader gets an up-to-date overview in the current field.

What I miss are figures and tables. The mammoth of written pages and text are just too heavy to cope with for a plain reader and he/she may lost interest in reading. I recommend adding tables (e.g. summarizing the references according to topic, and scope eg. coffee, brewing type, analyzed compound type etc.) thus reader will have a better overview and find instantly the information what he/she is looking for.

The same applies to figures. I would include a "flowchart" with some figures like a graphical description of the process of targeted and non-tartgered chapters both. And also possibly some chromatograms and mass spectra as example for identification and separation process if permission is available from cited authors.

Author Response

Reviewer 4:

Dear Authors and editors,

this is a comprehensive review of the targeted and non-targeted analysis of coffee and its products using LC and MS. The review is detailed enough, English language is OK as far as I can judge, the number of cited references is adequate for a review article and reader gets an up-to-date overview in the current field.

What I miss are figures and tables. The mammoth of written pages and text are just too heavy to cope with for a plain reader and he/she may lost interest in reading. I recommend adding tables (e.g. summarizing the references according to topic, and scope eg. coffee, brewing type, analyzed compound type etc.) thus reader will have a better overview and find instantly the information what he/she is looking for.

The same applies to figures. I would include a "flowchart" with some figures like a graphical description of the process of targeted and non-tartgered chapters both. And also possibly some chromatograms and mass spectra as example for identification and separation process if permission is available from cited authors.

Thank you for your advice. We have included two tables related to overview of targeted and non-targeted analysis. Further, we have added another 8 figures: a graphical description of targeted and non-targeted analysis, a graphical representation of their advantages and disadvantages, selected HPLC chromatograms related to both methodologies, and results of multivariate statistical methods used for coffee authenticity and quality assessment. Due to the limited revision time, we could attach only figures published in open access papers, thus not requiring copyright permissions.

Round 2

Reviewer 1 Report

The authors have followed the reviewers' suggestions and clearly improved the quality of the review manuscript. In my opinion, the review can be accepted in its current form.